# Response of Mediterranean Ornamental Plants to Drought Stress

**Stefania Toscano [1] , Antonio Ferrante [2] and Daniela Romano [1],***

1    Department of Agriculture, Food and Environment (Di3A), Università degli Studi di Catania,
     Via Valdisavoia 5, 95123 Catania, Italy; stefania.toscano@unict.it
2    Department of Agricultural and Environmental Sciences, Università degli Studi di Milano, Via Celoria 2,
     1-20133 Milano, Italy; antonio.ferrante@unimi.it
*    Correspondence: dromano@unict.it; Tel.: +39-095-234-306

**Abstract:** Ornamental plants use unique adaptive mechanisms to overcome the negative effects of drought stress. A large number of species grown in the Mediterranean area offer the opportunity to select some for ornamental purposes with the ability to adapt to drought conditions. The plants tolerant to drought stress show different adaptation mechanisms to overcome drought stress, including morphological, physiological, and biochemical modifications. These responses include increasing root/shoot ratio, growth reduction, leaf anatomy change, and reduction of leaf size and total leaf area to limit water loss and guarantee photosynthesis. In this review, the effect of drought stress on photosynthesis and chlorophyll *a* fluorescence is discussed. Recent information on the mechanisms of signal transduction and the development of drought tolerance in ornamental plants is provided. Finally, drought-induced oxidative stress is analyzed and discussed. The purpose of this review is to deepen our knowledge of how drought may modify the morphological and physiological characteristics of plants and reduce their aesthetic value—that is, the key parameter of assessment of ornamental plants.

**Keywords:** growth; gas exchange; chlorophyll fluorescence; oxidative stress; signal transduction; plant choice; green areas

## 1. Introduction

Drought stress strongly limits the growth of plants in Mediterranean regions. In the world, there are five Mediterranean-climate regions (i.e., areas surrounding the Mediterranean Sea, parts of western North America, parts of western and southern Australia, southwestern South Africa, and parts of central Chile) located between 32°–40° N and S of the Equator [1]. The Mediterranean climate is defined by precipitation and temperature, and it is characterized by a high seasonality summarized as hot and dry summers and cool and wet winters [2]. Despite the fact that these territories occupy less than 5% of the earth's surface, they harbor almost 20% of the world's vascular plant species [3]. The primary aspect that influences plant characteristics and natural vegetation is the extensive dry season. For this reason, plant growth and survival are endangered by long periods lacking rainfall and higher temperatures in the summer that impose more or less intense stress conditions [4]. The global climate changes that are occurring currently will worsen the availability of water, especially in arid and semi-arid environments. The availability of fresh and good quality water will decrease, especially in large cities [2,5]. This will entail difficulties in keeping green areas because the competition for water will be a critical issue.

For these reasons, great attention has recently been placed on the use and management of water to improve the sustainability of ornamental plant maintenance in semi-arid environments, such as

the Mediterranean basin. Water scarcity led to the diffusion of techniques for creating green spaces that are able to save water (xeriscaping), favoring the use of species tolerant of water stress, which are native species like the carob tree, a species that is highly tolerant to high temperature and to low soil water efficiency [6]. This attention to water saving depends on the fact that even if the water in the urban environment is widely used for purposes other than irrigation (for example industrial and residential uses), "a landscape may serve as a visual indicator of water use to the public due to its visual exposure" [7]. The water saving can be maximized by utilizing different strategies such as making a suitable choice of ornamental plant species—one that has a high tolerance to drought stress without compromising the ornamental value and/or reducing the effects of drought stress through innovative cultivation methods.

Ornamental plants are not only species and/or cultivars that offer aesthetic pleasure, but they can also improve the environment and the quality of our lives [8]. Thus, ornamental plants can be used to restore disturbed landscapes, control erosion, reduce energy for climatization and water consumption, and improve the aesthetic quality of urban, peri-urban, and rural landscapes, as well as recreational areas, interiors, and commercial sites. In consideration of the many contexts in which plant species can be used for ornamental purposes, the number is very large. The wide number of the ornamental or potentially ornamental species increases the possibility of finding suitable genotypes that are able to cope with drought stress and that can be used for landscaping planning.

For landscaping, plant choice can be based on a very large number of species from a wide geographical range and with different functions [8]. Unlike in agriculture, performance of an amenity landscape is not measured with a quantifiable yield, but rather how well it meets the expectations of the user or the individual paying for installation and maintenance. These expectations include aesthetic appearance and/or utility such as shading, ground cover, and recreation [9]. Sometimes, in degraded environments, plant survival is the only purpose of cultivation. Furthermore, for ornamental plants used in landscaping, fast growth is not always desirable because the excessive shoot vigor often requires frequent pruning with higher management costs. To maintain a compact growth habit, ornamental plants may have to be pruned or treated with plant growth regulators [10]. A reduced quantity of water may have positive benefits on growth control, therefore moderate drought stress can be a useful tool to provide plants with compact habit and slower growth—both parameters required for easier landscape management [11]. Plant drought stress is difficult to study because the sensitivities and response times to water deficit vary among different plant species and are related to the intensity and length of the water stress. Plant response to drought stress involves the interaction of various physiological and biochemical parameters that can be exploited as markers for the identification of tolerant species [12].

## 2. Ornamental Plant Response to Drought Stress

### 2.1. Growth and Morpho-Anatomical Modification

Plant responses to drought are different and interconnected. Plant plasticity to drought stress adaptation varies within genera, species [13,14], and even cultivars. The main morphological changes under drought conditions are shoot and leaf growth reduction. These negatively affect the ornamental value and the visual appearance, which are particularly important key factors (from the ornamental point of view) that must be used along with markers for selecting tolerant genotypes [15]. Several experimental studies on ornamental plants showed that plant quality decreased in response to severe drought stress [16–18].

The effect of drought stress on plant growth and dry matter has been noticed in numerous ornamental species—for example, *Pistacia* [12], *Spiraea, Pittosporum* [19], *Bougainvillea* [20], *Callistemon* [21], *Laurus,* and *Thunbergia* [22] (Table 1). Since the photosynthetic pathways strongly influence the response to water stress, only the responses of C3 plants are presented in Table 1.

The reduction of leaf area is another typical response observed in plants subjected to water stress, as confirmed by several authors. Indeed, as reported by Toscano et al. [23], the total leaf area and the leaf number showed the widest variations in *Lantana* between control and severe deficit irrigation, while in *Ligustrum,* the differences were more marked for the total leaf area and not significant for the leaf number. The reduction of the leaf area is a consequence of a reduction in the leaf number [24] or the leaf size (unit leaf area) [22]. Thus, plants counteract the water limitation by reducing the transpiration area. One of the avoidance mechanisms that minimizes water loss when the stomata are closed is, in fact, the reduction of the canopy area. In callistemon plants, drought stress increases the root-to-shoot ratio, causing the reduction of aerial tissues rather than the roots [25–27]. This reduction also occurs when the plants are grown in pots, a frequent condition in the nursery phase.

**Table 1.** Major effects of drought stress on ornamental plants [1].

| Species | Plant Habit | Treatments | Growth stage | Modified Parameter by Drought Stress | Ref. |
|---|---|---|---|---|---|
| *Rudbeckia hirta, Callistephus chinensis, Althaea rosea, Malva sylvestris* | forbs | 4 levels of irrigation treatments: 25%, 50%, 75% and 100% of the reference evapotranspiration (ET0) | Seedling one month after transplanting | Plant fresh weight (−); SLA (−); Stomatal Conductance (−); Δ Canopy Temperature (+); water use efficiency index (WUEi) (+); water use efficiency biomass (WUEb) (+) | [28] |
| *Periploca angustifolia* | bushy-branched shrub | Full irrigation (FI), Water Deficit (WD), and Rehydrated (R) | 11-month-old seedlings | Relative water content (RWC) (−); osmotic potential ($\psi\pi$) (−); water potential ($\psi$w) (−); transpiration rate (−); net $CO_2$ assimilation rate ($ACO_2$) (−); stomatal conductance ($g_s$) (−); water use efficiency (WUE) (+); Proline (+); MDA (+); chlorophyll (a, b, total and a/b) and carotenoid content (−); | [29] |
| *Pistacia lentiscus* | bushy shrub | C = 100% water holding capacity; Moderate Water irrigation (MW, 60% of the control) and Severe Water deficit (SW, 40% of the C) | 1-year-old seedlings | Dry weight (−), plant height (−), pre-dawn leaf water potential ($\Psi$l) (−); RWC (−) in SW | [12] |
| *Lantana camara, Ligustrum lucidum* | bushy shrubs | C = container capacity, or irrigated at 100% of water container capacity (WCC); light deficit irrigation (LDI), irrigated at 75% of WCC; moderate deficit irrigation (MDI), irrigated at 50% of WCC; and severe deficit irrigation (SDI), irrigated at 25% of WCC. | Two-month-old rooted cuttings | Dry weight (−); leaf number (−); total leaf area (−); leaf thickness (−); photosynthesis (−); stomatal conductance (−); variable to maximal fluorescence (Fv/Fm) (−); water potential (−). | [23] |
| *Bougainvillea buttiana* 'Rosenka' and *B.* 'Lindleyana' | shrubby vines | C = substrate moisture close to container capacity and irrigation applied when 20% of the water was leached; deficit irrigation (DI), 25% of the amount of water supplied in C. | Two-year-old plants | Leaf, flower, total biomass dry weight, total leaf area (−); stomatal resistance (+); $\Psi$l and $\Psi$p (+); Stomatal length and width (−) | [20] |
| *Spirea nipponica* (S), *Pittosporum eugenioides* (P), *Viburnum nudum* (V) | bushy shrubs | 4 irrigation levels (100, 70, 50, and 25% of container capacity) and Trinexapac-ethyl (TE) treatments (0.1, 0.2, and 0.3 L ha$^{-1}$) | Plant heights 10 (S and V) and 40 cm (P) | Leaf number and area (−), plant dry weight and height (−), root dry weight (+). A, E, and gs (−). The application of 0.2 and 0.3 L ha$^{-1}$ TE enhanced S, P and V tolerance to drought stress | [19] |
| *Acacia tortilis* subsp. *raddiana* | medium-sized tree | C = 80% of field capacity; Stress = withholding irrigation for 25 d. | 6-week-old seedlings | Leaf number (−), dry mass (−), shoot length and total leaf area (−), water potential (−), stomatal conductance (−); transpiration rates (−); chlorophyll fluorescence (−) only when soil WC was < 40%, soluble sugars (+). | [30] |
| *Viburnum opulus* and *Photinia X fraseri* 'Red Robin' | bushy shrubs | C = 100% ET; Moderate Water Deficit plants (MWD) received 60% ET and Severe Water Deficit (SWD) received 30% ET. | Plants grown in pots (24 cm in diameter) | Water potentials (−); Pn and $g_s$ (−) in SWD in *P.* x *fraseri*; $g_s$ and leaf transpiration (Tr) (−) in *V. opolus* | [31] |

**Table 1.** *Cont.*

| Species | Plant Habit | Treatments | Growth stage | Modified Parameter by Drought Stress | Ref. |
|---|---|---|---|---|---|
| *Callistemon laevis* | bushy shrub | Control (0.8 dS m$^{-1}$, 100% water holding capacity), WD (0.8 dS m$^{-1}$, 50% of the amount of water supplied in control), saline (4.0 dS m$^{-1}$, same amount of water supplied as control) and saline water deficit (4.0 dS m$^{-1}$, 50% of the water supplied in the control). | 2-year-old rooted cuttings | Total biomass (−); plant height (−); osmotic adjustment (−), leaf tissue elasticity (−) | [21] |
| *Viburnum opulus* L. and *Photinia x fraseri* 'Red Robin' | bushy shrubs | Control with 600 mLday$^{-1}$ (C), moderate WD (MWD) 66% of C and severe water deficit (SWD) received 33% of C. | One-year-old plants | Stem diameter (−). Modulus of elasticity (−) only in *Photinia* | [32] |
| *B. glabra* 'Sanderiana', *B. xbuttiana* 'Rosenka', *B.* 'Lindleyana' | shrubby vine | Three irrigation levels based on the daily water use 100% (C), 50% (MDI) or 25% (SDI) | Rooted cuttings | SDW (−), total DW (−), leaf number (−), leaf area (−), macronutrient concentration (−) in SDI. Stomatal resistance (+), leaf water potential (−), leaf osmotic potential. (−) | [33] |
| *Nerium oleander* | bushy shrub | C (field capacity); WD (withholding irrigation) | One-year-old plants | Stem elongation (−); Leaf FW (−); Leaf WC (−); Chl (a, b and total) (−); Proline (+); Glycine betaine (+); Total soluble sugar (+); Total phenolic compounds (+); Total flavonoids (+); ascorbate peroxidase (+); glutathione reductase (+). | [34] |
| *Callistemon citrinus* 'Firebrand' | bushy shrub | C (substrate moisture close to container capacity); moderate deficit irrigation (MDI) by applying 50% of the amount of C and severe deficit irrigation (SDI) by applying 25% of the C irrigation | 2-year-old rooted cuttings | RGR (−) in MDI; R/S ratio (+); WUE (+); $g_s$ in MDI and SDI (−); Pn/$g_s$ ratios (+); Stem water potential (−); Pn (−) in SDI | [24] |
| *Pelargonium x hortorum* | forb | C (100% of water field capacity = WFC); sustainable deficit irrigation (SDI), irrigated at 75% WFC throughout the experiment; regulated deficit irrigation I (RDI I), irrigated at 75% throughout the experiment, except during the flowering phase when plants were irrigated at 100%; regulated deficit irrigation II (RDI II), irrigated at 100% throughout the experiment, except during the flowering phase when plants were irrigated at 75%. | Rooted cuttings (4- to 5-cm tall and with 6–7 leaves) | Height (−), Flowering (−) RDI II; SDW (−), Number of leaves (−); Total leaf area (−). | [35] |
| *Eugenia uniflora* 'Etna Fire', *P. x fraseri* 'Red Robin' | bushy shrubs | Well-watered (WW), moderate drought stress (MD, 75%), severe drought stress (35%, SD). | Three months old rooted cuttings | A, $g_s$ and E (−); RWC (−); Fv/Fm (−); Proline and MDA (+) in Eugenia; MDA (+) in SD. | [36] |

**Table 1.** *Cont.*

| Species | Plant Habit | Treatments | Growth stage | Modified Parameter by Drought Stress | Ref. |
|---|---|---|---|---|---|
| *Myrtus communis* | bushy shrub | Control (C), 100% water holding capacity [leaching 15% (v/v) of the applied water;]; moderate water deficit; MWD, 60% of the C; severe water deficit; SWD,40% of the C. | Seedlings of 2-year-old | SDW ($-$); root dry weights ($-$), leaf numbers ($-$), Total leaf area ($-$), plant height ($-$) in SWD; plant height ($-$) in MDW. Root hydraulic resistance ($+$); leaf water potential pre-dawn ($-$); Pn ($-$). | [37] |
| *Pelargonium x hortorum* | forb | Control, C, container capacity; Moderate deficit irrigation, MDI, 60% of the C; Severe deficit irrigation, SDI 40% of C. After 2 months, all the plants were exposed to a recovery period of 15 days with the same irrigation regime applied to control plants, until the end of the experiment. | Rooted cuttings | SDW ($-$); leaf area ($-$); R/S ratio ($+$); Height ($-$); Width ($-$); $g_s$ ($-$); Pn ($-$). | [38] |
| *Callistemon citrinus, Laurus nobilis, Pittosporum tobira, Thunbergia erecta* | bushy shrubs | Two consecutive cycles of suspension/rewatering (S-R) compared with plants that were watered daily (C). | Six-month-old plants | SDW ($-$); R/S ratio ($+$); RWC ($-$); Leaf water potential ($-$), $g_s$ ($-$); Pn ($-$). | [22] |
| *Passiflora alata, P edulis, P. gibertii, P. setacea, P. cincinnata* | climbing vines | Two soil water regimes: soil field capacity and interruption of irrigation until the stomatal closure and apparent wilting of the whole plant. | Six month after sowing | Height ($-$); Dry weight of leaves, branches, roots ($-$); $g_s$ ($-$); palisade parenchyma thickness ($+$); leaf limb and spongy parenchyma thicknesses ($+$); stomatal diameter ($+$). | [39] |

[1] C = control; ET = Evapotranspiration; WD = Water deficit; SDW = shoot dry weight; ($-$) reduction due to WD; ($+$) increase due to WD.

Increased root-to-shoot ratios are frequently observed in plants under drought conditions which reduces water consumption [40] and increases water absorption [41]. This parameter is also suggested as a screening factor for grading plants with different stress tolerances. In addition to the reduction of the leaf area during drought stress, the modification of the leaf size and the cuticle thickness are also observed.

In a study conducted by Toscano et al. [23] on two ornamental shrubs (*Lantana* and *Ligustrum*) in a Mediterranean area, the analysis of leaf anatomical traits allowed the identification of the different strategies used during water stress conditions. During severe deficit irrigation, *Lantana* plants increased the spongy tissue rather than the palisade tissue; this anatomical modification facilitated the diffusion of $CO_2$ toward the fixation sites in order to increase the concentration gradient between internal air space and the atmosphere, thus enhancing the competition among cells for $CO_2$ and light [42]. In both species, an increase in the thickness of the spongy tissue and the palisade tissue was observed. The reduction of the specific leaf area could be a way to improve water use efficiency (WUE). In fact, thicker leaves usually have higher concentrations of chlorophylls and proteins per unit leaf area and thus have greater photosynthetic capacities per unit leaf area than thinner leaves [43].

The leaf anatomical modifications are species-specific. Thus, in *Polygala* and *Viburnum* plants subjected to four levels of irrigation treatments through the use of dielectric sensors (EC 5TE, Decagon Devices, Pullman, Washington, USA) to maintain the substrate water content (WS) equal to 10% (WC10%), 20% (WC20%), 30% (WC30%), and 40% (WC40% = control) of the pot volume, the leaf anatomical modifications were linked to spongy tissue in *Polygala* and palisade tissue in *Viburnum* (Figure 1).

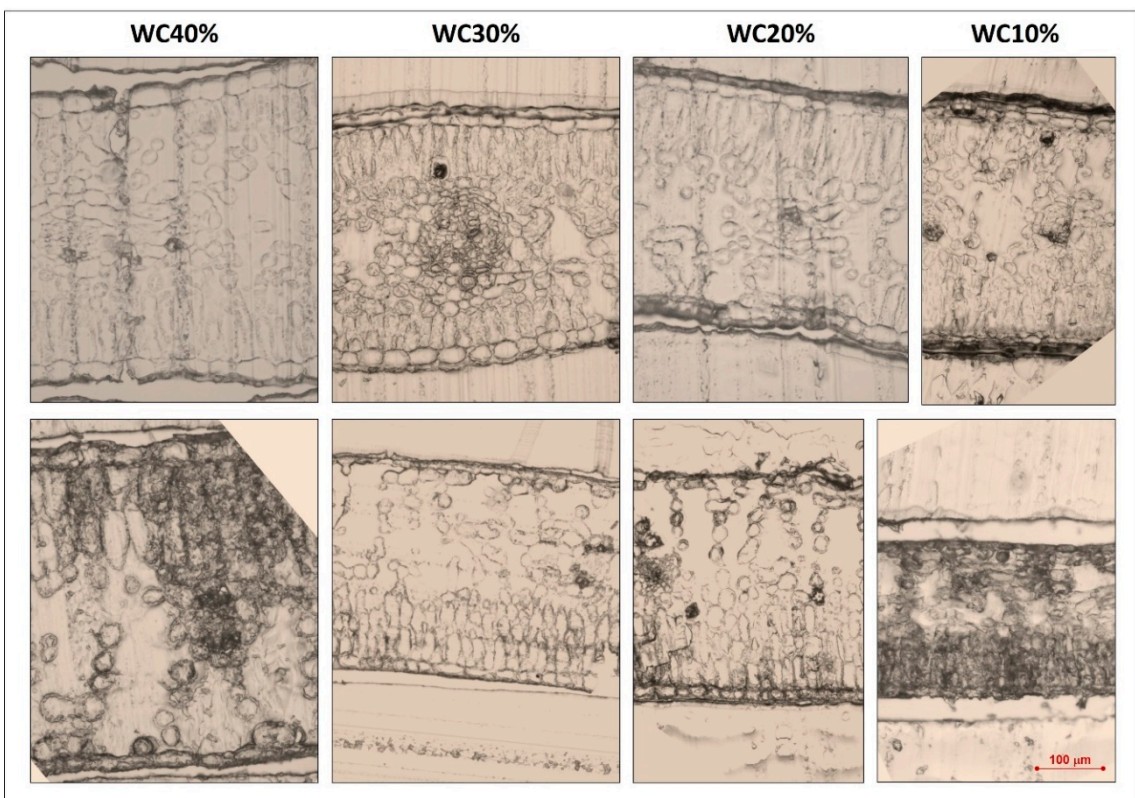

**Figure 1.** Light microscopy of blade cross-sections in *Polygala* (above) and *Viburnum* (below) at different water regimes (source: Toscano et al., unpublished data).

Acquiring greater knowledge of the morphological, physiological, and biochemical responses of the species in adverse environmental conditions, whether they are occasional, temporary, or long-term, allows us to choose the correct ornamental plants in relation to the interested area.

This information is useful in identifying the mechanisms of the adaptation of plants to adverse conditions such as drought stress [44], allowing us to select the most suitable species without compromising their aesthetic value.

*2.2. Physiological Parameters*

2.2.1. Leaf Gas Exchange

The main consequences of drought stress in plants are stomatal closure, reduction of gas exchange, the slowing down of photosynthetic activity, and the death of the plant [45,46]. Drought stress conditions mainly affect the photosynthetic system and ratio. In particular, they compromise the elements that are involved in the process, such as the electron transport to the thylakoids, the carbon cycle, and the stomatal control of $CO_2$ supply. Different published papers demonstrated that the reduction of photosynthetic activity is related to the mechanisms of stomatal conductance [47–50]. In fact, the first response of plants to water stress is stomatal closure and the subsequent reduction of the assimilation of the photosynthetic carbon necessary for the photosynthetic activity. As a consequence of the stomatal closure, there is not only a reduction in water loss, but also a reduction in nutrient uptake, consequently altering the metabolic pathways [51]. During drought stress, most species show a reduction in photosynthetic activity and a fast stomatal closure in relation to water potential adjustment [52,53]. The reduction in growth is also related to the reduction in the water potential of the leaves. Upon stomatal closure, a reduction in photosynthetic activity is achieved, which in turn leads to a decrease in plant growth and production [54,55]. The levels of carbon dioxide inside the stomatal chamber, and therefore in the cells, decrease, causing a reduction in photosynthesis. A decrease in the rate of $CO_2$ fixation is also observed and is associated with a reduction in the stomatal opening [56].

Under drought stress conditions, high conductivity ratios ($A_N$)/stomatal conductance ($g_s$) (also expressed as intrinsic WUE) indicate that leaves (the chloroplasts in particular)—even if there is an immediate stomatal closure—try to maintain high photosynthetic performance. As reported by Álvarez et al. [12], the decrease in $g_s$ in *Pistacia lentiscus* subjected to drought stress limited water losses through transpiration control.

In order to estimate the tolerance to drought stress in plants, the transpiration ratio is essential. In fact, it has been observed that species that can retain a greater quantity of water and therefore lose less water through the stomata are more tolerant to drought. [57]. As reported by Galmes et al. [58], shrubs have a better ability to regulate transpiration compared to herbaceous plants.

2.2.2. Chlorophyll *a* Fluorescence

Under water stress conditions, one parameter that is commonly used to identify the presence of photosynthetic plant damage in plants is the measurement of chlorophyll *a* fluorescence. In fact, this parameter is very useful for analyzing the influence of environmental factors on the efficiency of the photosynthetic apparatus [59]. Down regulation of photosystem II (PSII) activity results in an imbalance between the generation and utilization of electrons, apparently resulting in changes in quantum yield [60]. The ratio variable to maximal fluorescence ratio (Fv/Fm) (i.e., the maximum primary photochemical efficiency of the PSII in a sample of leaves adapted in the dark) allows the evaluation of the efficiency of the PSII photosystem, indirectly measuring the physiological state of the plant [61]. Several authors have defined the Fv/Fm threshold values to indicate if a plant is more or less stressed. Values between 0.78-0.85 indicate that the plant is not stressed [62]. In a study conducted by Álvarez et al. [63] on *Callistemon* plants maintained at different levels of drought stress, the Fv/Fm values remained constant at 0.80. The drought stress was not compromised by the PSII. Therefore, the *Callistemon* is a species resistant to drought. Álvarez et al. [12] reported that in *Pistacia lentiscus* plants subjected to different levels of water stress (from May to October), low Fv/Fm values were found in stressed plants during the warmer months. At the end of the trial when the conditions were less stressful, the plants recovered from these values. This shows that the plants did not cause

irreversible damage to the foliar tissues, indicating that PSII was not permanently damaged by stressful conditions. This affirms that the chloroplasts of Mediterranean species have different strategies during stress conditions for avoiding photo-inhibitory processes, such as the mechanism to consume the reducing power produced by the PSII [64,65].

## 2.3. Oxidative Stress

When photosynthetic activity is reduced and light excitation energy is in excess of that used or required by photosynthesis, over-excitation of the photosynthetic pigments in the antenna can occur, leading to the accumulation of reactive oxygen species (ROS) in chloroplasts [66]. During drought stress in plants, there are different biochemical changes. The main response is the accumulation of ROS, which causes the destruction of the cell membranes and results in oxidative damage to plants [67,68]. The plants, in order to oppose this accumulation, have developed many antioxidant activities and a series of secondary metabolites that counteract the generation of ROS and scavenge ROS once they are formed [69–71].

ROS are chemically active free radicals of oxygen. When unpaired electrons are present in the valence shell of these molecules, they become highly reactive and damage the cell structure and function. ROS production takes place within the compartments of different organelles, such as chloroplasts, mitochondria, and peroxisomes [60].

ROS include superoxide anion ($O^{2-}$), hydrogen peroxide ($H_2O_2$), hydroxyl radical ($OH^-$), singlet oxygen ($^1O_2$), and ozone ($O_3$). ROS are produced by plants continuously because they also have the role of cellular signaling, while excessive production involves oxidative stress [72].

Plants have mechanisms that protect them from the destructive action of oxidative reactions [73]. A mechanism put in place as a defense from stress relates to the production of antioxidant enzymes that protect the plants from ROS.

Garratt et al. [74] highlighted some enzymes among the main natural "detoxifiers" present in plants, such as superoxide dismutase (SOD; EC 1.15.1.1), catalase (CAT; EC 1.11.1.6), glutathione peroxidase (GPX; EC 1.11. 1.7), and ascorbate peroxidase (APX; EC 1.11.1.11). These enzymes are located in different compartments of the plant cells, while the CAT is instead located in the peroxisomes [75].

A type of ROS can be transformed into another type; for example, $O_3$ is decomposed into $H_2O_2$, $O^{2-}$, and $^1O^2$. The $O^{2-}$ is also transformed spontaneously or enzymatically into $H_2O_2$ through SOD activity [76], which can react further with $Fe^{2+}$ to form OH.

Controlling the production and action of ROS allows a better understanding of the effects of various abiotic stresses on plants. The study of protective mechanisms such as the antioxidant enzymes could allow the identification of processes that are the basis for the response of plants to stress.

When the plants are not stressed, the ROS level is kept low by the scavenger activity of the antioxidant enzymes. In the presence of abiotic or biotic stress (such as water, saline, or ozone stress), these balances are broken and there is an increase in the intracellular ROS levels. About 1%–3% of the oxygen that is consumed by plants leads to the formation of ROS [77,78]. The main changes that occur in plants are the increase in lipid peroxidation, protein degradation, DNA fragmentation, and finally cell death. All of this occurs because ROS are highly reactive [50]. Reacting with proteins and lipids, they modify structure, cellular metabolism, and, in particular, those that are linked to the photosynthetic process [79].

As a defense mechanism, the activity of these antioxidant enzymes increases under abiotic stress conditions such as drought [80–82], salinity [83,84], and ozone [85]. There are also non-enzymatic antioxidants: tocopherol, ascorbate, glutathione, phenols, alkaloids, flavonoids, and proline [60,72,86–91]. A decomposition product of poly-to-fatty acids of polyunsaturated fatty acids is malondialdehyde (MDA). It is considered a marker of membrane lipid peroxidation, which is an effect of oxidative damage. During the various drought stress conditions, some adapted species modify their antioxidant activities, increasing, for example, the activity of SOD and peroxidase (POD) [92]. SOD is the primary defense against ROS because it eliminates superoxide radicals. Specifically, it dismutates two $O_2^-$

radicals into $H_2O_2$ and $O_2^{-}$, which are precursors to other ROS and are generated in different subcellular compartments [93].

## 3. Mechanism of Signal Transduction and Development of Drought Tolerance

Drought stress is sensed by the roots of plants and the reduction of water availability slowly occurs depending on the soil physical properties. The limitation of water induces in plants several physiological, biochemical, and molecular changes that lead to increased plant tolerance (Figure 2). Since plants cannot escape from adverse weather conditions, survival depends on their ability to develop efficient adaptation strategies. The plant responses start from the activation of specific regulatory genes that lead to the modification of the physiology and the metabolism of the plants. Currently, transcriptional changes are widely studied in different species and under different drought stress conditions. Pioneer studies have been carried out on model plants, such as *Arabidopsis thaliana*, identifying the transcription profiles and transcription factors involved in responses to drought stress [94,95]. Among the different genes, dehydrin was found to be an indicator of the entire transcriptome response under drought stress. The increase in stress intensity induces the activation of genes associated with stress responses [96]. The most important genes involved belong to abscisic acid (ABA) perception and biosynthesis as well as the ethylene pathway. Among the transcription factors involved, the most important are abscisic acid-responsive element (ABRE), ABRE-binding (AREB) proteins, ABRE-binding factors (AREB/ABFs), drought-responsive *cis*-element binding protein/C-repeat-binding factor (DREB/CBF), ABF/AREB, NAC, WRKY transcription factors, Apetala 2 (AP2), and ethylene response elements [97,98]. The ABF/AREB are under ABA regulation involving SnRK2. These transcription factors are able to provide rapid gene activation under different abiotic stresses, including drought. Other transcription factors belong to the MYB family (such as MYB2 and MYC2) and are inducible by ABA [99]. Therefore, this plant hormone has a pivotal role under water stress in the activation of secondary gene networks, which leads to plant adaptation to stress. Mutants lacking ABA biosynthesis or action are very sensitive to drought stress [100].

The genes induced under drought encode for different proteins that are directly or indirectly involved in plant adaptation. Specific genes induced by water stress increase the accumulation of late embryogenesis abundant (LEA) proteins [101]. These proteins are accumulated in tissues under dehydration or desiccation, such as seeds. In plants, the LEA proteins are considered important in plant drought tolerance [102]. Water stress induces gene expression of membrane proteins. Among these, the most important are the aquaporins, i.e., the water channels.

At a biochemical level, plants increase the biosynthesis of osmolytes to lower the cell water potential and increase the water uptake ability of roots. These molecules are responsible for plant osmotic adaptation and include glycine, glycine betaine, proline, sugars, γ-aminobutyric acid, alcohols, sugar alcohols, trehalose, mannitol, polyamines, etc. [103–105]. The accumulation of these substances allows for the improvement of crop tolerance against drought stress, and the visual appearance of the plants does not change. Plants do not seem to be under stress conditions, but the biosynthesis of protectant molecules requires energy that is not exploited for the growth or the yield in agricultural crops. The energy used for the biosynthesis of osmolytes is also known as "fitness cost", which represents the energy costs for the plant to defend itself. The plants reduce photosynthetic activity, and ribulose-1,5-bisphosphate carboxylase/oxygenase (RUBISCO) efficiency declines with the increase in water reduction [106]. Since photosynthesis is a biochemical process that requires water, carbon dioxide, and light, the lack of water directly reduces photosynthesis. The quantum efficiency of PSII at the initial water stress transiently increases and then declines. The light received by the leaves must be dissipated to avoid photo-oxidative damage, and the energy dissipated can be estimated by chlorophyll *a* fluorescence. Gas exchange at the leaf level is regulated by stomatal opening. Under drought, water loss can be reduced by stomatal closure and a reduction in carbon dioxide concentration [107]. The reduction of light use can lead to an excess of excitation energy in leaves with ROS accumulation [108]. The increase in radicals stimulates the plant to activate the antioxidant

systems, such as the enzymes involved in the detoxification of cells. The most important enzymes are SOD, CAT, APX, POD, glutathione reductase (GR), and monodehydroascorbate reductase (MDHAR). These enzymes are able to reduce the ROS accumulation and enhance plant tolerance to drought [70]. Drought stress is a common stress in plants grown in the Mediterranean area, and several ornamental shrubs subjected to water availability increase the activity of these enzymes [36].

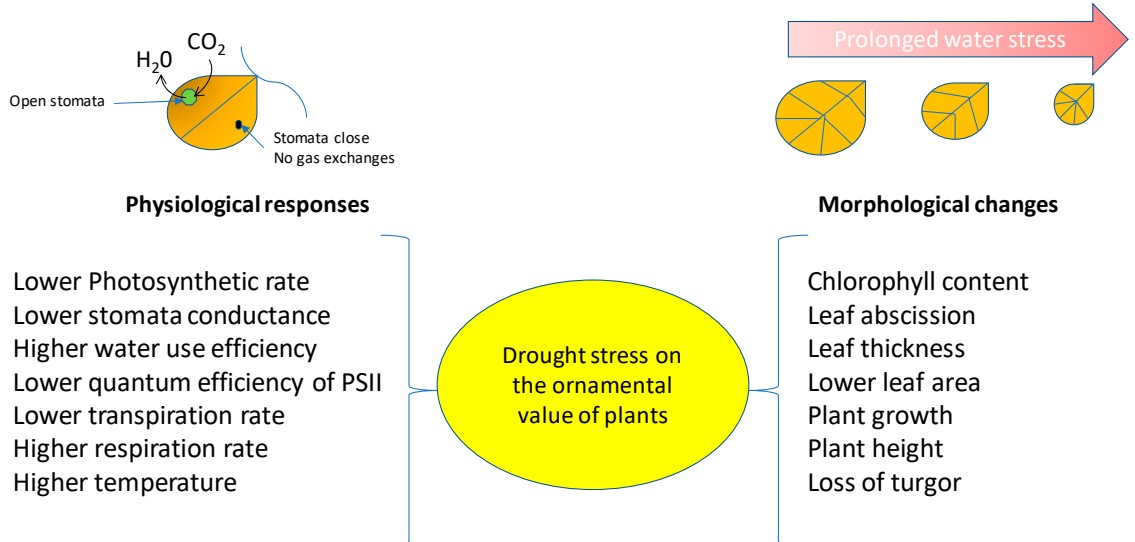

**Figure 2.** Physiological and morphological changes of plants exposed to reduced water availability. The magnitude of changes depends on the intensity of the stress.

Reduced photosynthetic activity also affects sugar concentration since respiration under drought increases because the plant temperature increases [109]. Plants under normal conditions are able to maintain the leaf temperature in the optimal range for photosynthesis by their thermoregulation ability, which is due to the evaporation of water at the leaf level through transpiration. The water passing from the aqueous state to the gas absorbs the heat from the plants and lowers the temperature. Under drought conditions, the closure of the stomata reduces transpiration and leads to a temperature increase, inducing a higher respiration rate. The lower photosynthesis and the higher respiration rate collectively reduce plant growth [110]. The reduction of plant growth in ornamental plants under water stress has been reported in several species, such as *Eugenia uniflora, Passiflora incarnata, and Photinia x* fraseri [36,111]. Ornamental plants can adopt different strategies under water stress. The study of plant responses to drought can be simulated by reducing water availability. In a study focused on drought responses, it was found that *Penstemon barbatus* was able to counteract drought by increasing root biomass and reducing stomatal conductance [112]. The gas exchange parameters, such as photosynthesis and stomatal conductance, can be considered good parameters for ornamental plant selection for tolerance to drought stress.

ABA is one of the most important plant hormones because it can regulate stomata opening in relation to potassium ions in guard cells [113]. An increase in ABA is crucial for reducing water loss through the stomata. Exogenous applications of ABA demonstrated that treated plants have a higher tolerance to drought. Another plant hormone that is induced by water stress is ethylene. It is also known as a senescence hormone because it is involved in leaf and flower senescence. Several ornamental plants are sensitive to ethylene, and it causes leaf abscission and yellowing, and petal rolling or desiccation [114]. Therefore, water stress can be detrimental for the ornamental plants used in the garden or other urban or peri-urban areas. Ethylene can be produced from endothermic engines. Therefore, in urban areas, plants exposed to ethylene and drought stress accelerate their senescence. Another important plant hormone that can have a positive role in the mitigation of drought stress is represented by the cytokinins. It has been demonstrated

that *Arabidopsis* plants overexpressing genes involved in cytokinin biosynthesis showed higher drought stress tolerance [115]. These plant hormones have a preferential site of biosynthesis in roots, and drought stress seems to reduce the concentration of cytokinins with an increase in root growth [116]. The increase in root biomass is considered a first response of the plant to drought stress. However, the application of some plant growth promoting bacteria (PGPB) also induced drought tolerance by increasing their cytokinin concentration and ABA [117].

Therefore, plant adaptation to drought stress is due to plant hormone equilibrium, and the plant responses are consequences of the cross-talk among them [118].

## 4. Effects of Drought Stress on the Ornamental Value of Plants

Plants under water stress modify their morphology and physiology to survive under stressful environments. These changes can also have a direct effect on the visual appearance and subsequently the ornamental value of the plants. Morphological changes can be observed on the leaves and the plant habit. The most common changes that are observed are leaves that are smaller and have different orientations on the branches. Ornamental plants used in drought-prone environments must be able to adapt to the utilization area, such as private gardens or urban or peri-urban areas without irrigation systems. At nursery levels, the selection for drought environment can be carried out by considering the size and architecture of the roots, which can explore a wide volume of soil. Unfortunately, evaluation of root systems is not easy to perform.

In nursery cultivation, the generalized use of pots, often of small volume, cause root restriction effects. Yong et al. [119] analyzed the influence of substrate volume reduction on cotton plants under conditions where water and nitrogen supplies were not limited. The root-restriction lowered the rate of photosynthesis due to lower stomatal conductance. Root restriction increased the shoot-to-root ratios and reduced the total whole-plant leaf area by 20%.

The critical step for many ornamental plants is transplanting. Therefore, the hardening of plants is important for xerophytic environments [120]. After transplanting, the survival of plants can be guaranteed from their ability to reduce water losses through transpiration and gas exchange. The adapted plants must reduce stomatal conductance, maintain their water balance, and have high WUE [121].

The effects of drought have direct impacts on the habit of plants, and the ornamental quality can be observed at the leaf level. Leaves can drop, change color, or show necrosis from the action of ethylene. Flower life and turnover are also affected in many ornamental plants. The presence of flowers on plants greatly enhances their visual appearance. Therefore, tolerant plants should be able to have a high number of flowers with longer lives because, under water deficits, the turnover of flowers is reduced [122]. Flower turnover or new flower production depends on plant energy availability. Under prolonged drought stress, reduced photosynthesis and fewer carbohydrates are available for flowering.

However, reduced growth can have positive effects for urban green areas and the maintenance of public and private gardens due to lower management costs. Reduced growth is particularly important with ornamental plants that are shaped by pruning. Slower growth contributes to a longer preservation of shape with delayed pruning activities.

## 5. Use of Different Tools in Mitigating Drought-Induced Damages

A solution to overcoming the problems associated with drought stress is making an appropriate plant choice. The response to drought varies greatly among the plants that can be used in landscaping. In green areas, often a combination of woody and herbaceous ornamental plants is used with various manufactured elements (generally referred to as 'hardscape') [123]. The plant choice can refer to a very large number of species in different environments that are able to assure different functions in the landscape [8]. Plant adaptability to drought stress changes within genera, species, or cultivars [13,14].

Where drought stress is frequent, the ornamental plant choice can favor plants that grow in desert areas (like xerophytes or succulents), which are especially capable of surviving water shortages.

Arbuscular mycorrhizal (AM) symbiosis can also increase host resistance to drought stress, although the effect is not always predictable. Since drought stress is frequent in drying soils, the AM influence on plant drought response can be the result of AM influence on salt stress. With this aim, Cho et al. [124] determined if the AM-induced effects on drought responses would be more accentuated when plants of similar sizes were exposed to drought in salinized soils, rather than only when drought was applied. In the trial, using two greenhouses, different water relations characteristics were measured in sorghum (*Sorghum bicolor*) plants colonized by *Glomus intraradices*, *Gigaspora margarita,* or a mixture of AM species during a sustained drought following exposure to salinity treatments (NaCl stress, osmotic stress via concentrated macronutrients, or soil leaching). The findings confirmed that AM fungi can alter the host response to drought, but they did not lend much support to the idea that AM induced salt resistance. The beneficial effects of AM were related to the improved ability of the roots to adsorb water by increasing the active root surface. The increase in the root adsorption ability was also due to gibberellin- and cytokinin-induced production by AM [125].

Direct and indirect positive roles of PGPB in plants under stress have been reported [126]. The positive effects of PGPB are through the activation of 1-aminocyclopropane-1-carboxylate deaminase enzyme that reduces ethylene production and increases auxin concentration in roots [127]. In recent years, there has been an increase in biostimulants used in agriculture and horticulture to enhance crop abiotic stress tolerance [128]. Alleviation of abiotic stress is perhaps the most frequently cited benefit of biostimulant formulations [129]. Biostimulants are derived from organic substances through different industrial processes. They can be composed of microorganisms such as fungi or bacteria [128] and help to improve plant abiotic adaptation by acting on the physiology and biochemistry of plants [130]. The cytokinin-producing bacteria under drought conditions are of relevant interest [131]. Some microbial inoculants known to have a positive effect on plant development can also help plants overcome or tolerate abiotic stress conditions. In ornamental plants, production can be improved by biostimulant application. Hibiscus (*Hibiscus* spp.) treated with commercial biostimulants showed an increase in gas exchange with higher photosynthetic activities [132]. In a pot experiment with bedding plants, a seaweed extract of *Ascophyllum nodosum* revealed positive effects on the growth and development of petunias (*Petunia* spp.), pansies (*Viola tricolor*), and cosmos (*Cosmos* spp.) exposed to drought [133]. Some reported positive effects of biostimulants are the induction of early flowering, a higher number of flowers, and higher biomass accumulation [134]. With the aim of evaluating the differences in the mechanisms involved in ornamental species' resistance to drought stress resulting from a regular suspension and recovery of the water supply, Toscano et al. [22] subjected plants of five ornamental shrubs (*Callistemon citrinus*, *Laurus nobilis*, *Pittosporum tobira*, *Thunbergia erecta,* and *Viburnum tinus* 'Lucidum') to two consecutive cycles of suspension/rewatering (S-R) and compared them with plants that were watered daily (C). The five species exhibited different responses to drought stress. At the end of the experimental period, S-R treatment had no effect on the dry weight of any species except *Pittosporum*. In *Pittosporum*, drought stress reduced total plant biomass by 19%. Drought stress induced alterations in shrubs, including decreases in shoot dry matter and increases in the root-to-shoot ratio, strongly affecting *Callistemon* and *Pittosporum*. All species adapted to water shortages using physiological mechanisms (RWC and water potential adjustment, stomatal closure, and reductions in photosynthesis). Following re-watering, the species fully recovered. Therefore, they can be considered as suitable for landscaping in the Mediterranean environment. However, *Laurus* and *Thunbergia* seemed to be less sensitive to drought stress than the other species.

Light drought stress can be adopted to control the growth of pot plants. Davies et al. [135] used deficit irrigation in comparison to conventional overhead irrigation in two crops of different canopy structure (*Cornus alba* and *Lonicera periclymenum*). In a subsequent experiment, *Forsythia* × *intermedia* was grown in two substrates with contrasting quantities of peat (60 and 100%). Deficit irrigation was found to be mainly effective in controlling vegetative growth when applied using overhead irrigation.

Similar results were achieved when drip irrigation was used. This comparable response suggests that deficit irrigation can be applied without precision drip irrigation. Scheduling two very different crops with respect to their water use and uptake potential, however, highlighted challenges in the application of appropriate deficits for very different crops under one system. Responses to deficit irrigation are more consistent where nursery management allows for scheduling of crops with very different architecture and water use under different regimes.

## 6. Conclusions and Future Prospective

The drought tolerance of ornamental plants widely varies with genotypes, environmental conditions, and soil or substrate characteristics. Landscape plants have similar mechanisms of drought tolerance to agricultural crops, but assessment of drought tolerance for these plants should be based primarily on aesthetic value rather than growth effects. Because of the wide number of plant species potentially available for ornamental purpose, it should be possible to choose genotypes suitable for drought environments.

Problems in research that occur are linked to: (i) the necessity to experimentally analyze a wide range of plant species to find those most suitable for specific sites; (ii) identifying parameters with simple measurements to discriminate tolerance to drought stress, and (iii) tailoring irrigation methods or plant management strategies to enable the chosen species to cope with water stress.

The study of the mechanism of plant response to drought stress and particularly of signal transduction and development of drought tolerance allow for the identification of suitable plants and management strategies for the cultivation or utilization of ornamental plants in drought-prone environments.

**Author Contributions:** D.R. and A.F. projected the design of the review. S.T. and D.R. wrote the introduction and the information related with growth and morpho-anatomical modification under drought stress; S.T. wrote all the information related with physiological parameters and oxidative stress; A.F. wrote all the information related with mechanism of signal transduction and development of drought tolerance. D.R. and S.T. wrote all the information related with effect of drought stress on ornamental value, the tool use for mitigating the drought. S.T. and D.R. made the Figure 1 and Table 1. AF made the Figure 2. D.R. and S.T. ordered all the references. All authors wrote the conclusions and revised and approved the manuscript.

**Funding:** This research received no external funding.

**Conflicts of Interest:** The authors declare no conflict of interest.

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
