# Peer review of "Response of Mediterranean Ornamental Plants to Drought Stress"

_horticulturae, doi:10.3390/horticulturae5010006_

Round 1
Reviewer 1 Report
The review concerns the responses of Mediterranean ornamental plants to specific climate conditions, especially to drought. The authors focus on morphological, physiological and biochemical changes that occur in plants in order to resist drought. Thus, an overall description of the mechanism against drought is presented. It is well structured and well written.
In some cases (eg page 9) references are lacking.
It can be said that a common tree used for amelioration of landscape in drought environments is Ceratonia siliqua L. This species is tolerant to high temperature and to low soil water efficiency (Ouzounidou et al. 2012). Carob shows an important capacity to control water loss, while climatic conditions prevailing in the Mediterranean basin do not threaten the survival of it.
The reference of: Ouzounidou G, Vekiari S, Asfi M, Gork M G, Sakcali M S, Ozturk M Photosynthetic characteristics of carob tree (Ceratonia siliqua L.) and chemical composition of its fruit on diurnal and seasonal basis. Pakistan Journal of Botany 44, 1689-1695, 2012, should be inserted.
The review after minor revision can be accepted for publication in the journal Horticulturae-MDPI.
Author Response
The review concerns the responses of Mediterranean ornamental plants to specific climate conditions, especially to drought. The authors focus on morphological, physiological and biochemical changes that occur in plants in order to resist drought. Thus, an overall description of the mechanism against drought is presented. It is well structured and well written. In some cases (eg page 9) references are lacking. It can be said that a common tree used for amelioration of landscape in drought environments is Ceratonia siliqua L. This species is tolerant to high temperature and to low soil water efficiency (Ouzounidou et al. 2012). Carob shows an important capacity to control water loss, while climatic conditions prevailing in the Mediterranean basin do not threaten the survival of it. The reference of: Ouzounidou G, Vekiari S, Asfi M, Gork M G, Sakcali M S, Ozturk M Photosynthetic characteristics of carob tree (Ceratonia siliqua L.) and chemical composition of its fruit on diurnal and seasonal basis. Pakistan Journal of Botany 44, 1689-1695, 2012, should be inserted.
The review after minor revision can be accepted for publication in the journal Horticulturae-MDPI
Authors Answer (A.A.): Thanks for the suggestion. We added more references according your comments in the paragraph “Mechanism of signal transduction and development of drought tolerance”. The citation on the carob was added in the introduction on your suggestion. The English has been revised.
Reviewer 2 Report
Please see attached

Author Response
Reviewer #2:
This is an interesting mini-review pertaining to the drought responses of a selected group of Mediterranean-like plants within a horticulture context.
The paper is adequately written and organised in a logical fashion in line with Horticulturae guidelines. The literature cited are relevant and updated.
Nevertheless, I have a few suggestions that are needed to improve the mini-review, before final acceptance.
Suggestions for improvements
Introduction
1. 28-46. The word “Mediterranean” should be interpreted broadly as several of the species (Acacia, Callistemon, etc ) highlighted here originated from a similar “Mediterranean”-like climate of Western Australia (Kwongan) and the Fynbos of South Africa. Perhaps the authors could consider explaining the dry summers and wet winters of a typical a “Mediterranean”-like environment.
A.A.: Thanks for your suggestion; we have added additional references in the introduction that explain the dry summers and wet winters in a Mediterranean environment.
.
2. 130-157
As the intrinsic drought tolerance ability is related to plant habit type (herbaceous, forb or shrub like and tree) and photosynthetic pathways (C3 vs C4 and CAM, and C3-CAM switching upon drought), Table 1 should include additional but relevant classification information of the species highlighted.
A.A.: In the table, we have added the plant habit type for each species. We have not added the photosynthetic pathways because are all C3 but has been specified in the paragraph "Growth and morpho-anatomical modification".
3. 310-336
As this mini-review is about drought within a horticultural context, the issue of water availability and therefore soil volume or pot size the authors should also consider adding a section about this important phenomenon (root restriction, pot sizes, etc.) affecting both water availability and the concomitant mineral nutrition constraints, ultimately affecting drought responses.
Yong, Wong, Letham and Farquhar (2010).
http://www.publish.csiro.au/FP/FP10009
Jackson (1993)
https://www.sciencedirect.com/science/article/pii/S0065229608602049
A.A.: We have modified the sentence according your suggestion. In particular, in nursery cultivation, the generalized use of pots, often of small volume, emphasize the root restriction effects. Yong et al. [2010] analyzed on cotton plants the influence of substrate volume reduction under conditions where water and nitrogen supplies were not limiting. The root-restriction lowers rates of photosynthesis, due to lower stomatal conductance. The root restriction increased the shoot-to-root ratios, and reduced the total whole-plant leaf area by 20%.
4. 346-355. As this paper specifically focussed on a phytohormone ABA (negative regulator; mentioned earlier in 300-309) that closes stomata, the authors should consider adding a few relevant references to explain the changes in the other known stomata-altering phytohormones (e.g. cytokinins) in response to the addition of beneficial microbes like AM.
Connecting to the earlier discussion of ABA, this begs the question of how beneficial microbes modulates and help the plants to cope with lesser water? In this respect, the authors did not provide a good connection, with relevant literature, as part of the solution to overcome drought conditions through harnessing a microbe-plant symbiotic strategy.
A.A.: The role of cytokinins under drought have been included in the text as suggested. The relationship between plant growth promoting bacteria and plant hormones have been reported and the reference suggested have been read and reported where appropriate.
5. Some suggestions for the authors to consider, and to strengthen the mini-review.
Involvement of phytohormones
https://www.frontiersin.org/articles/10.3389/fpls.2018.00655/full
Using bacteria to improve phytohormones
Glick (2014)
https://www.sciencedirect.com/science/article/pii/S094450131300150X
Liu et al (2013)
https://link.springer.com/article/10.1007%2Fs00253-013-5193-2
Using organic amendments to improve phytohormones and other factors
Wong et al (2015)
https://www.researchgate.net/publication/300238542_The_Importance_of_Phytohormones_and_Microbes_in_Biofertilizers
6. How do these phytohormones and other signals, counter the negative signals like ABA and ethylene, as part of a smart and effective homeostatic system to keep the plants alive when water availability is diminishing? The authors, in my view, should have some discussion of the drought adaptations solutions and also in finding plausible solutions to Figure 2.
A.A.: Thanks for your suggestion, we have modified and added some references in the paragraph according your suggestion.
The basic mechanisms behind the adaptation of plants to drought stress have been widely reviewed, this review was organised to be focused on ornamental plants and the effect of drought on growing and utilization. On these aspects, we wanted to provide an original contribution. The basic mechanisms involved in the plant adaptation have been reported as useful information for their practical application to the ornamental sectors. Therefore, the signalling pathways description and the relationship among plant hormones were not described in-depth in this review. However, we improved the discussion related to these aspects, but remaining oriented to practical application.
Round 2
Reviewer 2 Report
Please see attached

Author Response
Thanks for the corrections and suggestion. All punctual correction are made.
We also modified the paragraph (402-405) on biostimulants, according your suggestion.
Additional references are added. In particular: In the recent years there is an increase of biostimulants use in agriculture and horticulture to enhance the crops abiotic stress tolerance [128]. Alleviation of abiotic stress is perhaps the most frequently cited benefit of biostimulant formulations [129]. Biostimulants are derived from organic substances, through different industrial processes and they can be composed by microorganisms such as fungi or bacteria [128] and help to improve plant abiotic adaptation by acting on the physiology and biochemistry of plants [130]. The relevance of cytokinin-producing bacteria under drought conditions is a topic of relevant interest [131]. Some microbial inoculants known to have a positive effect on plant development can also help plants overcome or tolerate abiotic stress conditions. In ornamental plants production can be improved by biostimulants application. Hibiscus treated with commercial biostimulants showed an increase of gas exchanges with higher photosynthetic activities [132] In a pot experiment with bedding plants, seaweed extract of Ascophyllum nodosum were revealing positive effects on plant growth and development of petunia, pansy and cosmos exposed to drought [133]. Positive effect of biostimulants have been reported on inducing early flowering, higher number of flowers, and higher biomass accumulation [134].